# PCMC-NET:
# FEATURE-BASED PAIRWISE CHOICE MARKOV CHAINS

**Alix Lhéritier**
Amadeus SAS
F-06902 Sophia-Antipolis, France
`alix.lheritier@amadeus.com`

## ABSTRACT

Pairwise Choice Markov Chains (PCMC) have been recently introduced to overcome limitations of choice models based on traditional axioms unable to express empirical observations from modern behavior economics like context effects occurring when a choice between two options is altered by adding a third alternative. The inference approach that estimates the transition rates between each possible pair of alternatives via maximum likelihood suffers when the examples of each alternative are scarce and is inappropriate when new alternatives can be observed at test time. In this work, we propose an amortized inference approach for PCMC by embedding its definition into a neural network that represents transition rates as a function of the alternatives' and individual's features. We apply our construction to the complex case of airline itinerary booking where singletons are common (due to varying prices and individual-specific itineraries), and context effects and behaviors strongly dependent on market segments are observed. Experiments show our network significantly outperforming, in terms of prediction accuracy and logarithmic loss, feature engineered standard and latent class Multinomial Logit models as well as recent machine learning approaches.

## 1 INTRODUCTION

Choice modeling aims at finding statistical models capturing the human behavior when faced with a set of alternatives. Classical examples include consumer purchasing decisions, choices of schooling or employment, and commuter choices for modes of transportation among available options. Traditional models are based on different assumptions about human decision making, e.g. Thurstone's Case V model (Thurstone, 1927) or Bradley-Terry-Luce (BTL) model (Bradley & Terry, 1952). Nevertheless, in complex scenarios, like online shopping sessions presenting numerous alternatives to user-specific queries, these assumptions are often too restrictive to provide accurate predictions.

Formally, there is a universe of alternatives $U$, possibly infinite. In each choice situation, some finite *choice set* $S \subseteq U$ is considered. A *choice model* is a distribution over the alternatives of a given choice set $S$, where the probability of choosing the item $i$ among $S$ is denoted as $P_S(i)$. These models can be further parameterized by the alternatives' features and by those of the individual making the choice.

An important class of choice models is the Multinomial Logit (MNL), a generalization of the BTL model—defined for pairwise choices only—to larger sets. MNL models satisfy *Luce's axiom* also known as *independence of irrelevant alternatives* (Luce, 1959), which states that the probability of selecting one alternative over another from a set of many alternatives is not affected by the presence or absence of other alternatives in the set. Moreover, any model satisfying Luce's axiom is equivalent to some MNL model (Luce, 1977). Equivalently, the probability of choosing some item $i$ from a given set $S$ can be expressed as $P_S(i) = w_i / \sum_{j \in S} w_j$ where $w_i$ is the *latent value* of the item $i$. Luce's axiom implies *stochastic transitivity* i.e. if $P(a \rhd b) \geq 1/2$ and $P(b \rhd c) \geq 1/2$, then $P(a \rhd c) \geq \max\left(P(a \rhd b), P(b \rhd c)\right)$ where $P(i \rhd j) \equiv P_{\{i,j\}}(i)$ (Luce, 1977). Stochastic transitivity implies the necessity of a total order across all elements and also prevents from expressing cyclic preference situations like the stochastic rock-paper-scissors game described in Section 3.2. Thurstone's Case V model exhibits strict stochastic transitivity but does not satisfy Luce's axiom

(Adams & Messick, 1958). Luce's axiom and stochastic transitivity are strong assumptions that often do not hold for empirical choice data (see (Ragain & Ugander, 2016) and references therein). For example, Luce's axiom prevents models from expressing *context effects* like the *attraction effect* (also know as *asymmetric dominance* or *decoy* effect), the *similarity effect* and the *compromise effect*. The attraction effect occurs when two alternatives are augmented with an *asymmetrically dominated* one (i.e., a new option that is inferior in all aspects with respect to one option, but inferior in only some aspects and superior in other aspects with respect to the other option) and the probability of selecting the better, dominant alternative increases (Huber et al., 1982). The similarity effect arises from the introduction of an alternative that is similar to, and competitive with, one of the original alternatives, and causes a decrease in the probability of choosing the similar alternative (Tversky, 1972). The compromise effect occurs when there is an increase in the probability of choosing an alternative that becomes the intermediate option when a third extreme option is introduced (Simonson, 1989). Examples of these effects are visualized in Section 4.

A larger class of models is the one of Random Utility Models (RUM) (Block & Marschak, 1960; Manski, 1977), which includes MNL but also other models satisfying neither Luce's axiom nor stochastic transitivity. This class affiliates with each $i \in U$ a random variable $X_i$ and defines for each subset $S \subseteq U$ the probability $P_S(i) = P(X_i \geq X_j, \forall j \in S)$. RUM exhibits *regularity* i.e. if $A \subseteq B$ then $P_A(x) \geq P_B(x)$. Regularity also prevents models from expressing context effects (Huber et al., 1982). The class of Nested MNL (McFadden, 1980) allows to express RUM models but also others that do not obey regularity. Nevertheless, inference is practically difficult for Nested MNL models.

Recently, a more flexible class of models called Pairwise Choice Markov Chains has been introduced in Ragain & Ugander (2016). This class includes MNL but also other models that satisfy neither Luce's axiom, nor stochastic transitivity, nor regularity. This class defines the choice distribution as the stationary distribution of a continuous time Markov chain defined by some transition rate matrix. Still, it satisfies a weakened version of Luce's axiom called *uniform expansion* stating that if we add "copies" (with no preference between them), the probability of choosing one element of the copies is invariant to the number of copies. Although the flexibility of this class is appealing, the proposed inference is based on maximizing the likelihood of the rate matrix for the observed choices which is prone to overfitting when the number of observations for each possible alternative is small and is inappropriate when new alternatives can be seen at test time.

Alternatives and individuals making choices can be described by a set of features that can be then used to understand their impact on the choice probability. A linear-in-features MNL assumes that the latent value is given by a linear combination of the parameters of the alternatives and the individual. Features of the individual can be taken into account by these models but inference suffers from scarcity and is inappropriate when new alternatives can be seen at test time. The latent class MNL (LC-MNL) model (Greene & Hensher, 2003) takes into account individual heterogeneity by using a Bayesian mixture over different latent classes—whose number must be specified—in which homogeneity and linearity is assumed. A linear-in-features parameterization of PCMC, for features in $\mathbb{R}^d$, was suggested in (Ragain & Ugander, 2016, Appendix) but requires fitting a weight matrix of size $|U| \times d$, which makes it scale poorly and does not allow to predict unseen alternatives. In this work, we propose an *amortized inference* approach for PCMC in the sense that the statistical parameters are reused for any pair of alternatives and their number is thus independent of the size of the universe. In addition, we allow non-linear modeling by using a neural network.

In complex cases like airline itinerary choice, where the alternatives are strongly dependent on an individual-specific query and some features, like price, can be dynamic, the previous approaches have limited expressive power or are inappropriate. Two recently introduced methods allow complex feature handling for alternatives and individuals. Mottini & Acuna-Agost (2017) proposes a recurrent neural network method consisting in learning to point, within a sequence of alternatives, to the chosen one. This model is appealing because of its feature learning capability but neither its choice-theoretic properties have been studied nor its dependence on the order of the sequence. Lhéritier et al. (2019) proposes to train a Random Forest classifier to predict whether an alternative is going to be predicted or not independently of the rest of the alternatives of the choice set. This approach does not take into account the fact that in each choice set exactly one alternative is chosen. For this reason, the probabilities provided by the model are only used as scores to rank the alternatives, which can be interpreted as latent values—making it essentially equivalent to a non-linear MNL. To escape this limitation and make the latent values dependent on the choice set, relative features are added (e.g. the

price for $i$-th alternative price$_i$ is converted to price$_i / \min_{j \in S}$ price$_j$ ). The non-parametric nature of this model is appealing but its choice-theoretic properties have not been studied either.

In this work, we propose to enable PCMC with neural networks based feature handling, therefore enjoying both the good theoretical properties of PCMC and the complex feature handling of the previous neural network based and non-parametric methods. This neural network parameterization of PCMC makes the inference amortized allowing to handle large (and even infinite) size universes as shown in our experiments for airline itinerary choice modeling shown in Section 5.

## 2 BACKGROUND: PAIRWISE CHOICE MARKOV CHAINS

### 2.1 DEFINITION

A Pairwise Choice Markov Chain (PCMC) (Ragain & Ugander, 2016) defines the choice probability $P_S(i)$ as the probability mass on the alternative $i \in S$ of the stationary distribution of a continuous time Markov chain (CTMC) whose set of states corresponds to $S$. The model's parameters are the off-diagonal entries $q_{ij} \geq 0$ of a rate matrix $Q$ indexed by pairs of elements in $U$. Given a choice set $S$, the choice distribution is the stationary distribution of the continuous time Markov chain given by the matrix $Q_S$ obtained by restricting the rows and columns of $Q$ to elements in $S$ and setting $q_{ii} = -\sum_{j \in S \setminus i} q_{ij}$ for each $i \in S$. Therefore, the distribution $P_S$ is parameterized by the $|S|(|S|-1)$ transition rates of $Q_S$.

The constraint

$$q_{ij} + q_{ji} > 0 \tag{1}$$

is imposed in order to guarantee that the chain has a single closed communicating class which implies the existence and the unicity of the stationary distribution $\pi_S$ (see, e.g., Norris (1997)) obtained by solving

$$\begin{cases} \pi_S Q_S = \mathbf{0} \\ \pi_S \mathbf{1}^T = 1 \end{cases} \tag{2}$$

where $\mathbf{0}$ and $\mathbf{1}$ are row vectors of zeros and ones, respectively. Since any column of $Q_s$ is the opposite of the sum of the rest of the columns, it is equivalent to solve

$$\pi_S Q'_S = \begin{bmatrix} \mathbf{0} & | & 1 \end{bmatrix} \tag{3}$$

where $Q'_S \equiv \begin{bmatrix} ((Q_S)_{ij})_{1 \leq i \leq |S|, 1 \leq j < |S|} & | & \mathbf{1}^T \end{bmatrix}$.

### 2.2 PROPERTIES

In Ragain & Ugander (2016), it is shown that PCMC allow to represent any MNL model, but also models that are non-regular and do not satisfy stochastic transitivity (using the rock-scissor-paper example of Section 3.2).

In the classical red bus/blue bus example (see, e.g., Train (2009)), the color of the bus is irrelevant to the preference of the transportation mode "bus" with respect to the "car" mode. Nevertheless, MNL models reduce the probability of choosing the "car" mode when color variants of buses are added, which does not match empirical behavior. PCMC models allows to model this kind of situations thanks to a property termed *contractibility*, which intuitively means that we can "contract" subsets $A_i \subseteq U$ to a single "type" when the probability of choosing an element of $A_i$ is independent of the pairwise probabilities between elements within the subsets. Formally, a partition of $U$ into non-empty sets $A_1, \ldots, A_k$ is a *contractible partition* if $q_{a_i a_j} = \lambda_{ij}$ for all $a_i \in A_i, a_j \in A_j$ for some $\Lambda = \{\lambda_{ij}\}$ for $i, j \in \{1, \ldots, k\}$. Then, the following proposition is shown.

**Proposition 1** (Ragain & Ugander (2016)). *For a given $\Lambda$, let $A_1, \ldots, A_k$ be a contractible partition for two PCMC models on $U$ represented by $Q, Q'$ with stationary distributions $\pi, \pi'$. Then, for any $A_i$,*

$$\sum_{j \in A_i} P_U(j) = \sum_{j \in A_i} P'_U(j).$$

Then it is shown, that contractibility implies *uniform expansion* formally defined as follows.

**Definition 1** (Uniform Expansion). *Consider a choice between $n$ elements in a set $S^{(1)} = \{i_{11}, \ldots, i_{n1}\}$, and another choice from a set $S^{(k)}$ containing $k$ copies of each of the n elements: $S^{(k)} = \{i_{11}, \ldots, i_{1k}, i_{21}, \ldots, i_{2k}, \ldots, i_{n1}, \ldots, i_{nk}\}$. The axiom of uniform expansion states that for each $m \in \{1, \ldots, n\}$ and all $k \geq 1$,*

$$P_{S^{(1)}}(i_{m1}) = \sum_{j=1}^{k} P_{S^{(k)}}(i_{mj}).$$

### 2.3 INFERENCE

Given a dataset $\mathcal{D}$, the inference method proposed in Ragain & Ugander (2016) consists in maximizing the log likelihood of the rate matrix $Q$ indexed by $U$

$$\log \mathcal{L}(Q; \mathcal{D}) = \sum_{S \subseteq U} \sum_{i \in S} C_{iS}(\mathcal{D}) \log \left( P_S^Q(i) \right) \tag{4}$$

where $P_S^Q(i)$ denotes the probability that $i$ is selected from $S$ as a function of $Q$ and $C_{iS}(\mathcal{D})$ denotes the number of times in the data that $i$ was chosen out of set $S$.

This optimization is difficult since there is no general closed form expression for $P_S^Q(i)$ and the implicit definition also makes it difficult to derive gradients for $\log \mathcal{L}$ with respect to the parameters $q_{ij}$. The authors propose to use Sequential Least Squares Programming (SLSQP) to maximize $\log \mathcal{L}(Q; \mathcal{D})$, which is nonconcave in general. However, in their experiments, they encounter numerical instabilities leading to violations ($q_{ij} + q_{ji} = 0$) of the PCMC definition, which were solved with additive smoothing at the cost of some efficacy of the model. In addition, when the examples of each alternative are scarce like in the application of Section 5, this inference approach is prone to severe overfitting and is inappropriate to predict unseen alternatives. These two drawbacks motivate the amortized inference approach we introduce next.

## 3 PCMC-NET

We propose an amortized inference approach for PCMC based on a neural network architecture called PCMC-Net that uses the alternatives' and the individual's features to determine the transition rates and can be trained using standard stochastic gradient descent techniques.

### 3.1 ARCHITECTURE

**Input layer** For PCMC, the choice sets $S$ were defined as a set of indices. For PCMC-Net, since it is feature-based, the choice sets $S$ are defined as sets of tuples of features. Let $S_i$ be the tuple of features of the $i$-th alternative of the choice set $S$ belonging to a given feature space $\mathcal{F}_a$ and $I$ be the tuple of the individual's features belonging to a given feature space $\mathcal{F}_0$. The individual's features are allowed to be an empty tuple.

**Representation layer** The first layer is composed of a representation function for the alternatives' features

$$\rho_{w_a} : \mathcal{F}_a \to \mathbb{R}^{d_a} \tag{5}$$

and a representation function for the individual's features

$$\rho_{w_0} : \mathcal{F}_0 \to \mathbb{R}^{d_0} \tag{6}$$

where $w_0$ and $w_a$ are the sets of weights parameterizing them and $d_0, d_a \in \mathbb{N}$ are hyperparameters. These functions can include, e.g., embedding layers for categorical variables, a convolutional network for images or text, etc., depending on the inputs' types.

**Cartesian product layer** In order to build the transition rate matrix, all the pairs of different alternatives need to be considered, this is accomplished by computing the cartesian product

$$\{\rho_{w_a}(S_1), \ldots, \rho_{w_a}(S_{|S|})\} \times \{\rho_{w_a}(S_1), \ldots, \rho_{w_a}(S_{|S|})\}. \tag{7}$$

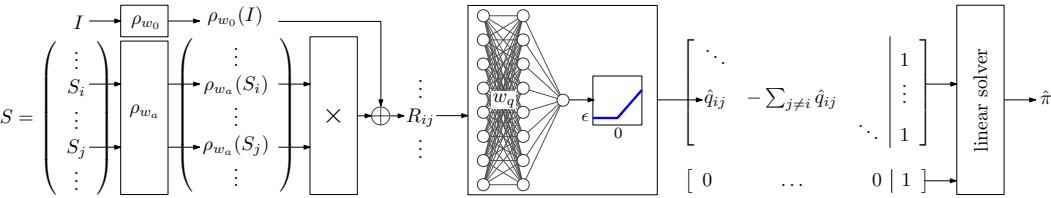

Figure 1: **PCMC-Net.** $\times$ denotes the cartesian product and $\oplus$ vector concatenation.

The combinations of embedded alternatives are concatenated together with the embedded features of the individual, i.e.

$$R_{ij} \equiv \rho_{w_0}(I) \oplus \rho_{w_a}(S_i) \oplus \rho_{w_a}(S_j) \tag{8}$$

where $\oplus$ denotes vector concatenation.

**Transition rate layer**     The core component is a model of the transition rate $(Q_S)_{ij}, i \neq j$:

$$\hat{q}_{ij} \equiv \max(0, f_{w_q}(R_{ij})) + \epsilon \tag{9}$$

where $f_{w_q}$ consists of multiple fully connected layers parameterized by a set of weights $w_q$ and $\epsilon > 0$ is a hyperparameter. Notice that taking the maximum with 0 and adding $\epsilon$ guarantees non-negativity and the condition of Eq. 1. The transition rate matrix $\hat{Q}$ is then obtained as follows:

$$\hat{Q}_{ij} \equiv \begin{cases} \hat{q}_{ij} & \text{if } i \neq j \\ -\sum_{j \neq i} \hat{q}_{ij} & \text{otherwise} \end{cases} . \tag{10}$$

**Stationary distribution layer**     The choice probabilities correspond to the stationary distribution $\hat{\pi}$ that is guaranteed to exist and be unique by the condition of Eq. 1 and can be obtained by solving the system

$$\hat{\pi} \left[ \left( \hat{Q}_{ij} \right)_{1 \leq i \leq |S|, 1 \leq j < |S|} \bigg| \mathbf{1}^T \right] = \left[ \mathbf{0} \mid 1 \right] \tag{11}$$

by, e.g., partially-pivoted LU decomposition which can be differentiated with automatic differentiation.

The whole network is represented in Fig. 1.

### 3.2    PROPERTIES

**Non-regularity**     As shown in Ragain & Ugander (2016), non-regular models can be obtained by certain rate matrices. For example, the stochastic rock-paper-scissors game can be described by a non-regular model obtained with the following transition rate matrix with $\frac{1}{2} < \alpha \leq 1$:

$$Q = \begin{bmatrix} -1 & 1-\alpha & \alpha \\ \alpha & -1 & 1-\alpha \\ 1-\alpha & \alpha & -1 \end{bmatrix} . \tag{12}$$

PCMC-Net can represent such a model by setting the following design parameters. In this case, the individual's features correspond to an empty tuple yielding an empty vector as representation. By setting $\rho_{w_a}$ to a one-hot representation of the alternative (thus $d_a = 3$), a fully connected network $f_{w_q}$ consisting of one neuron (i.e. six coefficients and one bias) is enough to represent this matrix since six combinations of inputs are of interest.

**Non-parametric limit**     More generally, the following theorem shows that any PCMC model can be arbitrarily well approximated by PCMC-Net.

**Theorem 1.** *If $\rho_{w_a}$ and $f_{w_q}$ are given enough capacity, PCMC-Net can approximate any PCMC model arbitrarily well.*

*Proof.* A PCMC model jointly specifies a family of distributions $\pi_S$ for each $S \in 2^U$ obtained by subsetting a single rate matrix $Q$ indexed by $U$. Therefore, it is sufficient to prove that PCMC-Net can approximate any matrix $Q$ since $\hat{q}_{ij}$, with $i \neq j$, does not depend on $S$. PCMC-Net forces the transition rates to be at least $\epsilon$, whereas the PCMC definition allows any $q_{ij} \geq 0$ as long as $q_{ij} + q_{ji} > 0$. Since multiplying all the entries of a rate matrix by some $c > 0$ does not affect the stationary distribution of the corresponding CTMC, let us consider, without loss of generality, an arbitrary PCMC model given by a transition rate matrix $Q^\star$, whose entries are either at least $\epsilon$ or zero. Let $\pi^\star$ be its stationary distribution. Then, let us consider the matrix $Q(\epsilon, c)$ obtained by replacing the null entries of $Q^\star$ by $\epsilon$ and by multiplying the non-null entries by some $c > 0$, and let $\pi(\epsilon, c(\delta))$ be its stationary distribution. Since, by Cramer's rule, the entries of the stationary distribution are continuous functions of the entries of the rate matrix, for any $\delta > 0$, there exist $c(\delta) > 0$ such that $|\pi(\epsilon, c(\delta)) - \pi^\star| < \delta$.

Since deep neural networks are universal function approximators (Hornik et al., 1989), PCMC-Net allows to represent arbitrarily well any $Q(\epsilon, c)$ if enough capacity is given to the network, which completes the proof. □

**Contractibility** Let $Q, Q'$ be the rate matrices obtained after the transition rate layer of two different PCMC-Nets on a finite universe of alternatives $U$. Then, Proposition 1 can be applied. Regarding uniform expansion, when copies are added to a choice set, their transition rates to the other elements of the choice set will be identical since they only depend on their features. Therefore, PCMC-Net allows uniform expansion.

## 3.3 INFERENCE

The logarithmic loss is used to assess the predicted choice distribution $\hat{\pi}$ given by the model parameterized by $w \equiv w_0 \cup w_a \cup w_q$ on the input $(I, S)$ against the index of actual choice $Y_S$,

$$\text{loss}(w, I, S, Y_S) \equiv \log \hat{\pi}_{Y_S}. \tag{13}$$

Training can be performed using stochastic gradient descent and dropout to avoid overfitting, which is stable unlike the original inference approach.

## 4 EXPERIMENTS ON SYNTHETIC DATA WITH CONTEXT EFFECTS

To illustrate the ability of PCMC and PCMC-Net models of capturing context effects we simulate them using the multiattribute linear ballistic accumulator (MLBA) model of Trueblood et al. (2014) that is able to represent attraction, similarity and compromise effects. MLBA models choice as a process where independent accumulators are associated to each alternative and race toward a threshold. The alternative whose accumulator reaches the threshold first is selected. The speed of each acculumator is determined by a number of parameters modeling human psychology notably, weights that determine the attention paid to each comparison and a curvature parameter determining a function that gives the subjective value of each alternative from its objective value.

We consider the example of (Trueblood et al., 2014, Figure 7) reproduced in Figure 2(i) where a choice set of two fixed alternatives $\{a, b\}$ is augmented with a third option $c$ and the preference of $a$ over $b$, i.e. $\frac{P_{\{a,b,c\}}(a)}{P_{\{a,b,c\}}(a) + P_{\{a,b,c\}}(b)}$, is computed. The model considers two attributes such that higher values are more preferable. The two original alternatives are $a = (4, 6)$, $b = (6, 4)$. For example, these can correspond to two different laptops with RAM capacity and battery life as attributes, and $c$ is a third laptop choice influencing the subjective value of $a$ with respect to $b$.

In order to generate synthetic choice sets, we uniformly sample the coordinates of $c$ in $[1, 9]^2$. Then, we compute the choice distribution given by the MLBA model[1] that is used to sample the choice.

We instantiate PCMC-Net with an identity representation layer with $d_a = 4$ and $d_0 = 0$ and a transition rate layer with $h \in \{1, 2, 3\}$ hidden layers of $\nu = 16$ nodes with Leaky ReLU activation

---

[1]Using the code available at `https://github.com/tkngch/choice-models`.

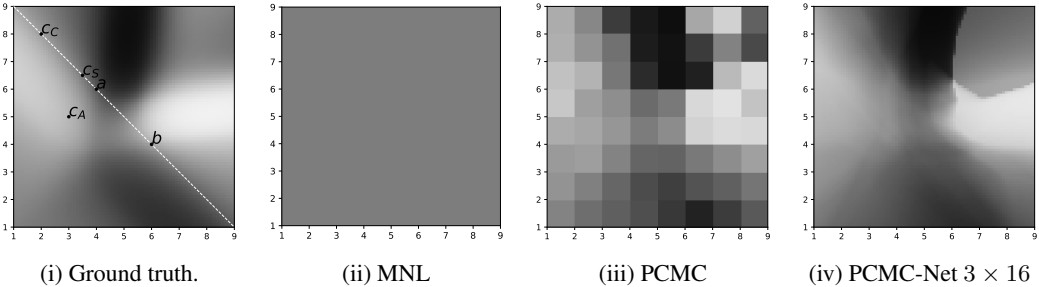

|     (i) Ground truth.     |     (ii) MNL     |     (iii) PCMC     |     (iv) PCMC-Net $3 \times 16$     |

Figure 2: Preference for $a$ over $b$ for different attributes (coordinates of each pixel) of the third alternative $c$. Lighter shades indicate higher preference for $a$. Models were trained on 20000 choice sets. In Fig. (i), the points $c_A$, $c_S$ and $c_C$ show examples of, respectively, attraction, similarity and compromise effects with respect to $a$.

(slope $= 0.01$) and $\epsilon = 0.5$.[2] In order to use the original PCMC model[3], we discretize the attributes of the third option using 8 bins on each attribute, obtaining 64 different alternatives in addition to identified $a$ and $b$. We also compare to a linear-in-features MNL model[4]. Figure 2 shows how the different models represent the preference for $a$ over $b$. Table 1, shows a Monte Carlo estimation of the expected Kullback-Leibler divergence comparing each model $\hat{P}$ to the true MLBA model $P$

$$\mathbb{E}_{c \sim \mathcal{U}([1,9]^2)}[D_{\mathrm{KL}}(P \| \hat{P})] = \frac{1}{64} \int_{[1,9]^2} \sum_{i \in \{a,b,c\}} P_{\{a,b,c\}}(i) \log \frac{P_{\{a,b,c\}}(i)}{\hat{P}_{\{a,b,c\}}(i)} dc. \tag{14}$$

As expected, MNL is unable to represent context effects due the independence of irrelevant alternatives. After discretization, the original PCMC provides a good approximation of the MLBA model. Nevertheless, it is difficult to further refine it since the number of statistical parameters grows biquadratically with the number of bins for each feature. As shown in Table 1, the amortized inference approach of PCMC-Net allows a better approximation of the original model with significantly fewer statistical parameters than the discretized PCMC.

Table 1: Monte Carlo estimate of the expected KL divergence between the different models and the true MLBA model. A set of 10000 points $c$, different from the training one, was used.

| $\hat{P}$ | $h \times \nu$ | #parameters | $\mathbb{E}_{c \sim \mathcal{U}([1,9]^2)}[D_{\mathrm{KL}}(P \| \hat{P})]$ |
|---|---|---|---|
| MNL |  | 2 | .119 |
| PCMC |  | 4290 | .022 |
| PCMC-Net | $1 \times 16$ | 97 | .018 |
| PCMC-Net | $2 \times 16$ | 369 | .011 |
| PCMC-Net | $3 \times 16$ | 641 | .009 |

## 5 EXPERIMENTS ON AIRLINE ITINERARY CHOICE MODELING

In this section, we instantiate PCMC-Net for the case of airline itinerary choice modeling. As shown in Babutsidze et al. (2019), this kind of data often exhibit attraction effects, calling for more flexible models such as PCMC. Nevertheless, in the considered dataset, alternatives rarely repeat themselves, which makes the original inference approach for PCMC inappropriate.

---

[2]We trained it with the Adam optimizer with one choice set per iteration, a learning rate of 0.001 and no dropout, for 100 epochs. Code is available at `https://github.com/alherit/PCMC-Net`.

[3]We used the implementation available at `https://github.com/sragain/pcmc-nips`, with additive smoothing $\alpha = 0.1$ and a maximum of 50 iterations. The best model over 20 restarts was considered.

[4]Using Larch available at `https://github.com/jpn--/larch`.

## 5.1 DATASET

We used the dataset from Mottini & Acuna-Agost (2017) consisting of flight bookings sessions on a set of European origins and destinations. Each booking session contains up to 50 different itineraries, one of which has been booked by the customer. There are 815559 distinct alternatives among which 84% are singletons and 99% are observed at most seven times. In total, there are 33951 choice sessions of which 27160 were used for training and 6791 for testing. The dataset has a total of 13 features, both numerical and categorical, corresponding to individuals and alternatives (see Table 4).

## 5.2 INSTANTIATION OF PCMC-NET

PCMC-Net was implemented in PyTorch (Paszke et al., 2017)[5] During training, a mini-batch is composed of a number of sessions whose number of alternatives can be variable. Dynamic computation graphs are required in order to adapt to the varying session size. Stochastic gradient optimization is performed with Adam (Kingma & Ba, 2015). In our experiments, numerical variables are unidimensional and thus are not embedded. They were standardized during a preprocessing step. Each categorical input of cardinality $c_i$ is passed through an embedding layer, such that the resulting dimension is obtained by the rule of thumb $d_i := \min(\lceil c_i/2 \rceil, 50)$. We maximize regularization by using a dropout probability of $0.5$ (see, e.g., Baldi & Sadowski (2013)). The additive constant $\epsilon$ was set to $0.5$. The linear solver was implemented with `torch.solve`, which uses LU decomposition. Table 2 shows the hyperparameters and learning parameters that were optimized by performing 25 iterations of Bayesian optimization (using GPyOpt authors (2016)). Early stopping is performed during training if no significant improvement (greater than $0.01$ with respect to the best log loss obtained so far) is made on a validation set (a random sample consisting of 10% of the choice sessions from the training set) during 5 epochs. Using the hyperparameters values returned by the Bayesian optimization procedure and the number of epochs at early stopping (66), the final model is obtained by training on the union of the training and validation sets.

Table 2: Hyperparameters optimized with Bayesian optimization.

| parameter | range | best value |
|---|---:|---:|
| learning rate | $\{10^{-i}\}_{i=1\ldots6}$ | 0.001 |
| batch size (in sessions) | $\{2^i\}_{i=0\ldots4}$ | 16 |
| hidden layers in $f_{w_q}$ | $\{1, 2, 3\}$ | 2 |
| nodes per layer in $f_{w_q}$ | $\{2^i\}_{i=5\ldots9}$ | 512 |
| activation | $\{$ReLU, Sigmoid, Tanh, LeakyReLU$\}$ | LeakyReLU |

## 5.3 RESULTS

We compare the performance of the PCMC-Net instantiation against three simple baselines:

- Uniform: probabilities are assigned uniformly to each alternative.
- Cheapest (non-probabilistic): alternatives are ranked by increasing price.
- Shortest (non-probabilistic): alternatives are ranked by increasing trip duration.

We also compare against the results presented in Lhéritier et al. (2019)

- Multinomial Logit (MNL): choice probabilities are determined from the alternatives' features only, using some feature transformations to improve the performance.
- Latent Class Multinomial Logit (LC-MNL): in addition to the alternatives' features, it uses individual's features which are used to model the probability of belonging to some latent classes whose number is determined using the Akaike Information Criterion. Feature transformations are also used to improve the performance.
- Random Forest (RF): a classifier is trained on the alternatives as if they were independent, considering both individual's and alternatives' features and using as label whether each alternative was chosen or not. Some alternatives' features are transformed to make them relative to the values of each choice set. Since the classifier evaluates each alternative

---

[5]Code available at `https://github.com/alherit/PCMC-Net`.

independently, the probabilities within a given session generally do not add to one, and therefore are just interpreted as scores to rank the alternatives.

And, finally, we compare to

- Deep Pointer Networks (DPN) (Mottini & Acuna-Agost, 2017): a recurrent neural network that uses both the features of the individual and those of the alternatives to learn to point to the chosen alternative from the choice sets given as sequences. The results are dependent on the order of the alternatives, which was taken as in the original paper, that is, as they were shown to the user.

We compute the following performance measures on the test set $\mathcal{T}$ of choice sets and corresponding individuals:

- Normalized Log Loss (NLL): given a probabilistic choice model $\hat{P}$,
  $\text{NLL} \equiv -\frac{1}{|\mathcal{T}|} \sum_{(S,I) \in \mathcal{T}} \log \hat{P}_S(Y_S|I)$.
- TOP $N$ accuracy: proportion of choice sessions where the actual choice was within the top $N$ ranked alternatives. In case of ties, they are randomly broken. We consider $N \in \{1, 5\}$.

Table 3 shows that PCMC-Net outperforms all the contenders in all the considered metrics. It achieves a 21.3% increase in TOP-1 accuracy and a 12.8% decrease in NLL with respect to the best contender for each metric. In particular, we observe that the best in TOP $N$ accuracy among the contenders are LC-MNL and RF, both requiring manual feature engineering to achieve such performances whereas PCMC-Net automatically learns the best representations. We also observe that our results are significantly better than those obtained with the previous deep learning approach DPN, showing the importance of the PCMC definition in our deep learning approach to model the complex behaviors observed in airline itinerary choice data.

Table 3: Results on airline itinerary choice prediction. * indicates cases with feature engineering.

| method | TOP 1 | TOP 5 | NLL |
|---|---|---|---|
| Uniform | .063 | .255 | 3.24 |
| Cheapest | .164 | .471 | – |
| Shortest | .154 | .472 | – |
| MNL* | .224 | .624 | 2.44 |
| LC-MNL* | .271 | .672 | 2.33 |
| RF* | .273 | .674 | – |
| DPN | .257 | .665 | 2.33 |
| PCMC-Net | **.331** | **.745** | **2.03** |

## 6 CONCLUSIONS

We proposed PCMC-Net, a generic neural network architecture equipping PCMC choice models with amortized and automatic differentiation based inference using alternatives' features. As a side benefit, the construction allows to condition the probabilities on the individual's features. We showed that PCMC-net is able to approximate any PCMC model arbitrarily well and, thus, maintains the flexibility (e.g., allowing to represent non-regular models) and the desired property of uniform expansion. Being neural network based, PCMC-Net allows complex feature handling as previous machine learning and deep learning based approaches but with the additional theoretical guarantees.

We proposed a practical implementation showing the benefits of the construction on the challenging problem of airline itinerary choice prediction, where attraction effects are often observed and where alternatives rarely appear more than once—making the original inference approach for PCMC inappropriate.

As future work, we foresee investigating the application of PCMC-Net on data with complex features (e.g. images, texts, graphs ...) to assess the impact of such information on preferences and choice.

ACKNOWLEDGMENTS

Thanks to María Zuluaga, Eoin Thomas, Nicolas Bondoux and Rodrigo Acuña-Agost for their insightful comments and to the four anonymous reviewers whose suggestions have greatly improved this manuscript.

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

## A    FEATURES OF THE AIRLINE ITINERARY CHOICE DATASET

Table 4: Features of the airline itinerary choice dataset.

|  | Type | Feature | Range/Cardinality |
|---|---|---|---|
| Individual | Categorical | Origin/Destination | 97 |
|  |  | Search Office | 11 |
|  | Numerical | Departure weekday | [0,6] |
|  |  | Stay Saturday | [0,1] |
|  |  | Continental Trip | [0,1] |
|  |  | Domestic Trip | [0,1] |
|  |  | Days to departure | [0, 343] |
| Alternative | Categorical | Airline (of first flight) | 63 |
|  | Numerical | Price | [77.15,16781.5] |
|  |  | Stay duration (minutes) | [121,434000] |
|  |  | Trip duration (minutes) | [105, 4314] |
|  |  | Number connections | [2,6] |
|  |  | Number airlines | [1,4] |
|  |  | Outbound departure time (in s from midnight) | [0, 84000] |
|  |  | Outbound arrival time (in s from midnight) | [0, 84000] |

