# OpenReview forum: "PCMC-Net: Feature-based Pairwise Choice Markov Chains"
_ICLR.cc/2020/Conference — Accept (Poster)_

### Official Review · AnonReviewer2 · 2019-10-21
**Official Blind Review #2**

**Rating:** 3

**Review:**

This paper presents an approach for choice modeling that leverages neural network features in a continuous-time Markov Chain whose stationary distribution represents the choice distribution. Nonlinear features can be computed for both alternatives and individuals, and the resulting model beats all baselines in terms of log-likelihood and accuracy on an airline itinerary choice prediction task.

Overall, this paper presented a simple but effective approach for using neural networks in the PCMC class of models. The experimental section is too limited, with results on only one dataset and no comparison of different architectural choices for how to incorporate neural networks into PCMC models, or analysis pointing toward what the features are learning that allows them to improve over earlier approaches. The text was also confusing in a number of places (possibly due to my lack of knowledge in choice modeling), and there’s no discussion of related work incorporating neural networks into ranking-based models.

Comments:
* Knowing nothing about choice modeling, I found the introduction hard to follow with lots of jargon that may be inaccessible to the broader ML community. It may be useful to specify the set of desired properties for these models up front, and then highlight how the different existing models do or don’t satisfy these properties (e.g. uniform expansion, regularity, efficiency,  framing effects, etc.)
* How is the proposed approach “amortized inference”?
* What’s the triangle notation in P(a |> c) ?
* It’d be useful to say more as to why contractability/uniform expansion are useful components of a choice model.
* “Additive smoothing at the cost of some efficacy” - efficacy in what sense? Expressivity? Or worse performance? The same smoothing technique (minimum of epsilon) seems to be used in this approach.
* Theorem 1 and proof do not consider some of the architectural choices of PCMC-Net (e.g. cartesian product layers, does d_a have to go to infinity?)
* Contractability: why is this property desirable if you don’t take advantage of it computationally?
* Why is SGD + dropout training stable but the original MLE problem not?
* Comparison to baselines using low-rank or other simple parameterizations of Q?
* Table 1 should be moved to supplement
* “Usual rule of thumb” -> cite something here? Not familiar with this
* What is the impact of \epsilon and dropout probability on model performance? From proofs I had expected \epsilon tiny (1e-9) but you use 0.5
* Would be useful to show how model performs on smaller dataset to gain intuition

**Experience Assessment:**

I do not know much about this area.

**Review Assessment: Checking Correctness Of Derivations And Theory:**

I assessed the sensibility of the derivations and theory.

**Review Assessment: Checking Correctness Of Experiments:**

I assessed the sensibility of the experiments.

**Review Assessment: Thoroughness In Paper Reading:**

I read the paper at least twice and used my best judgement in assessing the paper.

---

> ### Author Response · Authors · 2019-11-15
> **Response**
>
> Thank you for your comments and suggestions.
> 	• Regarding the jargon: we added a paragraph explaining the different types of effects and a new experiment on a synthetic dataset illustrating them--see new Section 4---.
> 	• Amortized inference: the same parameters are used to determine the transition rates between any pair of alternatives (even unseen ones) in contrast to the original PCMC approach that requires one statistical parameter for each possible pair of alternatives of the universe. We added a sentence in the introduction to explicitly define it.
> 	• The triangle notation is defined in the sentence where it is used first.
> 	• The original inference approach counts the choices observed for each choice set (see Eq. 4).  Additive smooting adds a pseudo-count (e.g. \alpha=0.1) to each counter to avoid numerical instabilities when doing the MLE optimization. The remark was taken from the original paper and refers to performance. The epsilon used in our work is used to enforce the constraint that defines PCMC, ensuring the existence and unicity of the stationary distribution and has no effect on performance.
> 	• Contractibility guarantees that the model behaves well when alternatives can be partitioned into classes of equivalence. This has practical modeling importance when equivalent alternatives are present in the choice set like in the classical Red Bus/Blue Bus example where the color of the bus is irrelevant to the preference of transportation mode "bus" with respect to the "car" mode. For example, MNL models reduce the probability of choosing the car mode when multiple color variants of buses are added, which does not match empirical behavior.  This example is added in Section 2.
> 	• In the new experiment, we compared against a simple parameterization of Q obtained by the discretizing the features.
> 	• We moved Table 1 to appendix
> 	• The new experiment considers a small synthetic dataset where present effects are known in order to gain intuition.

---

### Official Review · AnonReviewer1 · 2019-10-23
**Official Blind Review #1**

**Rating:** 6

**Review:**

This paper introduces a novel approximate inference method, called PCMC-Net, for models from the family of Pairwise Choice Markov Chains (PCMC). The method relies on training a neural network. Consequently, the authors claim that inference is amortized, but its computational complexity is still quadratic in the number of choice alternatives due to separate processing of all pairs of alternatives. PCMC-Net bakes the definition of PCMC into the neural net structure and therefore satisfies the theoretical properties of contractability and uniform expansion, which are desired properties of choice models
Moreover, since choice probabilities are a function of choice candidates’ features (and features of an individual making the choice), this method allows for new (unseen) choice candidates at test time, which was not possible with previously proposed maximum-likelihood (ML) inference. The approach is evaluated on modelling the choice of airline itinerary, on which it outperforms all considered baselines by a significant margin.

I recommend REJECTing this paper. This paper tackles the problem of efficient inference and test-time generalization (to unseen choice alternatives) for choice modelling, and the proposed approach is interesting, seems to be theoretically sound, and outperforms evaluated baselines. Experimental evaluation is insufficient, however, with the method assessed only on a single dataset---in which case it is unclear if the method is better than baselines in general, or whether it is a quirk of the considered dataset. Moreover, the authors do not compare to ML inference in PCMC, which seems to be the closest possible baseline; instead, the authors only mention that ML would overfit on this dataset. Finally, the paper is full of complicated terms and cumbersome notation, which makes it difficult to read. Technical terms are often used without definition (e.g. framing effects, Luce’s axiom, asymmetric dominance), which makes the paper inaccessible to an inexperienced reader like myself.

I think that this work could be improved in the following ways. The exposition should be made simpler and easier to follow (especially section 2), and all technical terms should be appropriately defined. Additionally, the method should be evaluated on at least one more dataset and compared to ML inference for PCMC. I am happy to increase my score if (all) the above points are addressed.


**Experience Assessment:**

I do not know much about this area.

**Review Assessment: Checking Correctness Of Derivations And Theory:**

I assessed the sensibility of the derivations and theory.

**Review Assessment: Checking Correctness Of Experiments:**

I carefully checked the experiments.

**Review Assessment: Thoroughness In Paper Reading:**

I read the paper at least twice and used my best judgement in assessing the paper.

---

> ### Author Response · Authors · 2019-11-15
> **Response**
>
> Thank you for your comments and suggestions.
> Regarding the technical terms: framing effect was replaced by context effect which is more standard and is now properly defined in the introduction along with the other choice-theoretic terms. We added an example to motivate the definition of contractibility in Section 2.
> We added a new experiment---see new Section 4--- that uses a different dataset that illustrates these context effects and compares to the original PCMC.
> Although it is possible to compare to the original PCMC by resorting to a discretization of the feature space, it quickly becomes impractical as the number of attributes and the number of bins grow, making it inappropriate on a complex dataset such as the airline itinerary choice one, which is the point of proposing the PCMC-Net approach.

---

### Official Review · AnonReviewer3 · 2019-10-24
**Official Blind Review #3**

**Rating:** 6

**Review:**

Summary:  This paper enables a feature-based parametrization and amortized inference of Pairwise Choice Markov Chains (PCMCs), a model for decisions in the face of a set of alternative choices (e.g. the rock-paper-scissors game).  Previous approaches to fitting PCMCs have leveraged sequential least squares programming, making optimization unstable, the model prone to overfitting, and test-time inference difficult.  The authors propose parametrizing PCMCs with neural networks to fix these issues.  Relying on universal function approximation results, the authors show that their PCMC-Net can represent arbitrary transition matrices.  The experiments report results on a dataset of airline booking behavior, comparing PCMC-Net with four other baselines from the literature.

Pros:  Although I was previously unfamiliar with the PCMC model, using a neural network parametrization seems novel and well motivated.  Moreover, the airline data experiment seems to validate that PCMC-Net is indeed effective, besting the other baselines in all three metrics.

Cons:  I have two main concerns over the paper’s experimental rigor…

#1 Lack of Simulation Studies: The paper makes claims about the representation properties of PCMC-Net but fails to validate them with simulation studies.

#2 Lack of data sets: Only one experiment on one data set is reported.

Final Evaluation:  While I found the paper’s methodology well motivated and sensible, the experiments do not thoroughly validate the method as they contain no simulation studies and only one data set.


**Experience Assessment:**

I do not know much about this area.

**Review Assessment: Checking Correctness Of Derivations And Theory:**

I assessed the sensibility of the derivations and theory.

**Review Assessment: Checking Correctness Of Experiments:**

I assessed the sensibility of the experiments.

**Review Assessment: Thoroughness In Paper Reading:**

I read the paper at least twice and used my best judgement in assessing the paper.

---

> ### Author Response · Authors · 2019-11-15
> **Response**
>
> Thank you for your comments and suggestions.
> We added a new experiment on a synthetic dataset---see new Section 4---to show the ability of PCMC-Net to represent context effects (whose definition was added in the introduction).

---

> > ### Comment · AnonReviewer3 · 2019-11-15
> > **Re Response**
> >
> > Thanks for adding the experiment.  I have raised my score to 'weak accept.'

---

### Official Review · AnonReviewer4 · 2019-10-31
**Official Blind Review #4**

**Rating:** 8

**Review:**

1.The goal of the paper is to connect flexible choice modeling with a modern approach to ML architecture to make said choice modeling scalable, tractable, and practical.
2. The approach of the paper is well motivated intuitively, but could more explicitly show that PCMC-Net is needed to fix inferential problems with PCMC and that e.g. SGD and some regularization + the linear parameterization suggested by the original PCMC authors isn't scalable in itself.
3. The approximation theorem is useful and clean, and the empirical results are intriguing. While consideration of more datasets would improve the results, the metrics and baselines considered demonstrate a considerable empirical case for this method.

My "weak accept" decision is closer to "accept" than "weak reject." (Edit 11/25/19: I raised my score in to accept in conjunction with the author's improvements in the open discussion phase)

Improvement areas(all relatively minor):
- While I personally enjoy the choice axioms focused on by the PCMC model and this paper, stochastic transitivity, IIA, and regularity are probably more important to emphasize than Contractibility. Because the properties of UE and contractibility were not used, it may be more appropriate to use this space to introduce more of the literature on neural-nets-as-feature-embeddings stuff.
- This paper could be improved by generalizing to a few other choice models- in particular the CDM (https://arxiv.org/abs/1902.03266) may be a good candidate for your method. This is more a suggestion for future work if you expand this promising initial result.
- Hyper-parameter tuning: I noticed that several of your hyper parameters were set to extremal values for the ranges you considered. If you tuned the other algorithms' hyper parameters the same way, it could be the case that the relative performance is explained by the appropriateness of those ranges. Would be interesting to have a more in-depth treatment of this, but I do understand that it's a lot of work.


Specific Notes:
Theorem 1 is nice, and the proof is clean, but doesn't explicitly note that a PCMC model jointly specifies a family of distributions \pi_S for each S \in 2^U obtained by subsetting a single rate matrix Q indexed by U. It's clear that PCMC-Net will still approximate under this definition, as \hat q_ij  approximates each q_ij because \hat q_ij doesn't depend on S. While the more explicit statement is true with the same logic in the theorem, the notational choice to have "X_i" represent the "i-th" element in S is confusing at first, as e.g. X_1 is a different feature vector for S = {2,3} and S={1,3}. I don't see this issue as disqualifying, but it took me a while to realize that there wasn't more than a notational abuse problem when I returned to the definitions where the indexing depended on the set S under consideration.


Typos/small concerns:
-Above equation (1), the number of parameters in Q_S is |S|(|S|-1) rather than (|S|-1)^2, as each of the |S| alternatives has a transition rate to the other |S|-1 alternatives.
-Below equation (3), I think you mean j \in S rather than 1<= j <= |S|, as S may not be {1,2,...,|S|}. Later I noticed that you always index S with {1,\dots,|S|}, but using i \in S in combination with 1<=j<=|S| was a bit confusing.
-X_i as the i-th element of S is a bit of an abuse of notation, as it surpasses dependence on S
-In Figure 1, you show X_0 in a vector that is referred to as "S." It is my understanding that X_0 represents user features. As the user is not in the set, this is confusing. The use of a vertical ellipsis to connect \rho(X_0) to \rho(X_1) is also confusion, as \rho(X_1) is input into the Cartesian product while X_0 is input into the direct sum.

Overall, nice job! Really enjoyed the paper and approach, good to see connections made between these literatures so that progress in discrete choice can be used at scale.


**Experience Assessment:**

I have published in this field for several years.

**Review Assessment: Checking Correctness Of Derivations And Theory:**

I carefully checked the derivations and theory.

**Review Assessment: Checking Correctness Of Experiments:**

I assessed the sensibility of the experiments.

**Review Assessment: Thoroughness In Paper Reading:**

I read the paper at least twice and used my best judgement in assessing the paper.

---

> ### Author Response · Authors · 2019-11-15
> **Response**
>
> Thank you for your comments and suggestions.
>
> 2. : we made this explicit in the introduction where we mention the linear approach suggested by the original PCMC authors.
> 3. : we added experiments on a synthetic dataset---see new Section 4.
>
> Thanks for the CDM suggestion, we will consider it for future work.
> Regarding hyper-parameter tuning of the competitors: we used the same dataset and the same hyper-parameters suggested by the authors of the respective papers who performed numerous experiments.
>
> Specific Notes:
> Thanks for pointing this out, we added the remark at the beginning of the proof.
>
> To make the notation more explicit regarding the dependence on S, we replaced X_i by S_i. We also replaced X_0 by I.
> We also added a sentence explicitly saying that for PCMC-Net, S is a set of tuples.
>
> Typos/Small concerns:
> 	- We fixed the wrong number of parameters
> 	- Below equation 3, the indices are with respect to Q_S and range from 1 to |S|, so the wrong part is i\in S. Therefore, we changed the indices to 1 <= i <= |S|, 1 <= j < |S| (the last inequality is a strict one since we want to remove the last column of Q_S. We fixed equation 11 in the same way.

---

### Decision · Program_Chairs · 2019-12-19

**Decision:**

Accept (Poster)

**Comment:**

This submission proposes to use neural networks in combination with pairwise choice markov chain models for choice modelling. The deep network is used to parametrize the PCMC and in so doing improve generalization and inference.

Strengths:
The formulation and theoretical justifications are convincing.
The improvements are non-trivial and the approach is novel.

Weaknesses:
The text was not always easy to follow.
The experimental validation is too limited initially. This was addressed during the discussion by adding an additional experiment.

All reviewers recommend acceptance.